

# 1  21st Century Asian air pollution impacts glacier in northwestern
# 2  Tibet

M. Roxana Sierra-Hernández[1], Emilie Beaudon[1], Paolo Gabrielli[1,2], and Lonnie G. Thompson[1,2]
[1]Byrd Polar and Climate Research Center, The Ohio State University, Columbus, OH, 43210, USA
[2]School of Earth Sciences, The Ohio State University, Columbus, OH, 43210, USA
*Correspondence to*: M. Roxana Sierra-Hernández (sierra-hernandez.1@osu.edu)
**Abstract.** Over the last four decades, Asian countries have undergone significant economic development leading to
rapid urbanization and industrialization in the region. Consequently, fossil fuel consumption has risen dramatically
worsening the air quality in Asia. Fossil fuel combustion emits particulate matter containing toxic metals that can
adversely affect living organisms, including humans. Thus, it is imperative to investigate the temporal and spatial
extent of metal pollution in Asia. Recently, we reported a continuous and high-resolution 1650–1991 ice core record
from the Guliya ice cap in northwestern Tibet, China showing a contamination of Cd, Pb and Zn during the 20th
century. Here, we present a new continuous and high-resolution ice core record of trace metals from the Guliya ice
cap that comprises the years between 1971 and 2015, extending the 1650–1991 ice core record into the 21st century.
Non-crustal Cd, Pb, Zn and Ni enrichments increased since the 1990s reaching a maximum in 2008. The enrichments
of Cd, Pb, Zn, and Ni increased by ~75 %, 35 %, 30 %, and 10 %, respectively during the 2000–2015 period relative
to 1971–1990. Our analysis suggests that emissions from Pakistan's fossil fuel combustion (by road transportation
and the manufacturing and construction industries) became the dominant source of Cd, Pb, Zn, and Ni deposited on
Guliya between 1995 and 2015. However, it is possible that emissions from Central Asia, Afghanistan, India, Nepal,
and the Xinjiang province in China have also impacted Guliya during the 21st century. The enrichments of Cd, Zn,
and Ni declined after 2008 likely due to a coal consumption decrease in Pakistan at that time. This new record
demonstrates that the current emissions in Asia are impacting remote high-altitude glaciers in the region and that
mitigation policies and technologies should be enforced to improve the air quality as economic development continues
in most Asian countries.

## 27  1 Introduction

Atmospheric trace elements (TE), including toxic metals (e.g., Hg, Pb, Cd) have dramatically increased since the 19th
century due to human activities (Pacyna and Pacyna, 2001; Tian et al., 2015). Some TEs are highly toxic and harmful
to an array of animals, plants and humans. Atmospheric TEs can originate from natural sources/processes in the
environment such as windborne dust, wildfires, sea-spray aerosols, volcanic activity, and from vegetation (Nriagu,
1989a, b). TEs are also released into the atmosphere by human activities including: 1) the combustion of fossil fuels
including coal, oil and its distillates (e.g., gasoline, jet fuel, diesel); 2) industrial processes such as mineral, and metal



production; 3) agriculture practices that include the use of fertilizers and pesticides; and 4) waste disposal (Pacyna
and Pacyna, 2001).
Various anthropogenic sources, including fossil fuel combustion and smelting, emit fine ($< 2.5$ µm) particulate matter
($PM_{2.5}$) which can contain toxic metals such as As, Cd, Pb, and Zn. Due to its small size, the lifetime of $PM_{2.5}$ in the
atmosphere can last for over a week, thereby allowing it to be transported and deposited far from its initial sources
(e.g., onto remote glaciers) (Pacyna and Pacyna, 2001; Marx and McGowan, 2010).
Since the 1980s, Asian countries such as China, India, Pakistan, Nepal, Bangladesh and others have undergone
significant and rapid economic growth, leading to considerable urbanization and industrialization in the region.
Consequently, fossil fuel combustion has risen dramatically in most of these countries worsening the air quality. In
particular, China and India, the second and fifth largest economies in the world according to the International Monetary
Fund, are respectively the largest and third largest emitters of both $CO_2$ and $PM_{2.5}$.
The rapid economic growth of China has been the result of its economic reform and open policy beginning in 1978
after Mao Zedong's death and the subsequent five-year plans (FYP) implemented by the Chinese government. With
their economic growth, China and India started to increase their coal consumption since ~1970 amplifying the global
atmospheric emissions of $CO_2$ and $PM_{2.5}$ (Crippa et al., 2018; EDGARv4.3.2, 2017).
In 1999, the Chinese government implemented the "Western Development" policy in the 10th FYP to improve the
quality of the environment in the east and to transfer energy (West to East energy program) and mineral resources
from the west to the rest of the country (Chen et al., 2010; Lai, 2002; Dong and Yang, 2014). For this purpose, the
necessary infrastructure (e.g., airports, railways, highways, water infrastructure, power lines) was built. As a result,
energy consumption (Jianxin, 2016) and atmospheric emissions (Liu et al., 2015) have been increasing significantly
in western China.
In particular, the Xinjiang Uygur Autonomous Region, situated in an arid region of Northwestern China, has become
important for the Western Development implementation because of its location on the new Silk Road and its large
reserves of oil, gas, and coal (Fridley et al., 2017; Chen et al., 2010; Dong and Yang, 2014). Three mountain ranges
shape the topography of this province: the Altai on the northern border, Tien Shan in the center, and the western
Kunlun Mountains, where the Guliya ice cap is located (see below), along the southern border with Tibet (Fig. 1a).
Pakistan, to the southeast of Xinjiang, gained its independence in 1947 after which its population rose very rapidly
becoming the world's sixth most populated country by 2003 (UN, 2017). Although Pakistan's economic growth has
been much lower than China's and India's, urban Pakistani cities are among the most polluted in the world (WHO,
2016) due to the high population growth, industrialization and a significant increase of motor vehicles that lack
emission controls and use low quality gasoline/diesel. The number of registered vehicles increased by ~400 % between
1996 (3.838 million) and 2014 (15.168 million) (Bajwa, 2015). Emissions from motor vehicles (60–70 %), industry,
and the generation of ~54,000 t of solid waste per day which is either dumped or incinerated have been estimated as
the principal sources of $PM_{2.5}$ and air pollution in Pakistan (Colbeck et al., 2010; Sánchez-Triana et al., 2014).



For the year 1995, Pacyna and Pacyna (2001) estimated that non-ferrous metal production was the largest source of As, Cd, Cu, In, and Zn; while coal combustion was the major source of Cr, Hg, Mn, Sb, Se, Sn, and Tl, and oil combustion was the major source of Ni and V, both worldwide and in Asia. For the same year, these authors estimated that leaded gasoline was the major source of Pb worldwide as well as in Asia. However, changes in emissions of atmospheric TEs have occurred during the 21st century in Asia due to the following: 1) China and India emerged as the fastest growing economies and most populated countries in the world (UN, 2017); 2) developing countries such as Pakistan, Nepal and Bangladesh have significantly increased their national economic activities since the 1980s–1990s; 3) temporal and regional variations in the implementation of control emission technologies and air quality standards (often higher concentrations than those recommended by the World Health Organization); 4) leaded gasoline was banned in 2000 in China, India, and Nepal and in 2002 in Pakistan, while it is still consumed in Afghanistan. Therefore, it is imperative to study the spatial and temporal effects of these new pollution sources and their resulting impacts on the environment.

Atmospheric emissions estimates are associated with large uncertainties due to inaccurate statistical information, the lack of field data and limited temporal and spatial coverage of observations. Thus, natural registers of past environmental conditions such as glaciers, which are influenced only by deposition of atmospheric species, are essential for reconstructing time series of atmospheric metal depositions (Hansson et al., 2015; Cooke and Bindler, 2015; Gabrielli and Vallelonga, 2015; Marx et al., 2016) that can be further used by modelers to reconstruct past emissions and project future atmospheric contamination trends.

Recently, we obtained a 350-year (1650–1991) high-resolution TE record (Sierra-Hernández et al., 2018) using an ice core drilled in 1992 from the Guliya ice cap located in northwestern China, specifically in Tibet's Kunlun Mountains (35° 17.37' N; 81° 29.73' E; 6200 m a.s.l.) (Thompson et al., 1995) (Fig. 1a). Outside of the Arctic and Antarctica, the glaciers in the Kunlun Mountains, along with those in Tibet and the Himalayas, are the largest reservoir of ice on the globe and are commonly referred to as the "Third Pole". This glacial region is the source for numerous rivers in Asia which provide water to hundreds of millions of people. The 1992 Guliya TEs record showed that long-distance emissions from coal combustion in Europe were likely deposited on the ice cap between 1850 and 1940 (Sierra-Hernández, 2018). Additionally, Pb, Cd, and Sn enrichments were detected between 1975 and 1991. The origin of these more recent enrichments could not be determined as more anthropogenic sources have emerged, especially in this region.

Here, we use a new ice core retrieved from Guliya in 2015 to extend the 1650–1991 TEs record into the 21st century (1971–2015) and to determine the impacts of the recent emission changes in Asia on the western Kunlun Mountains glaciers. This study fills a temporal and spatial gap in the investigation of atmospheric toxic trace metals in Northwestern China where atmospheric emission data are limited.


**2 Methodology**
**2.1 Guliya cores**
In 2015, two ice cores (309.73 m and 72.40 m long) were extracted from the plateau of the Guliya ice cap (6200 m
a.s.l), in close proximity to the 1992 drilling site. The timescale was constructed by annual layer counting using three
fixed horizons at 2015 (surface of the glacier), at 1992 corresponding to the surface of the 1992 core (at 6 m in the
shallow core), and at 1963 (at 10.9 m in the shallow core determined by beta radioactivity from the Arctic
thermonuclear tests). Annual layers were determined using both cores by matching the signals of at least three different
parameters ($Cl^-$ and $Na^+$, dust and $Ca^{2+}$, $SO_4^{2-}$, and $\delta^{18}O$). Dating uncertainties are estimated at 1–2 years between the
fixed points and may be the result of the very low annual accumulation (~230 mm water equivalent) and surface-
alteration processes such as snow redistribution through wind. Details of the drilling operation, the ice cores, and the
timescale can be found in Thompson et al. (2018).
**2.2 Sample preparation and ICP-SFMS analysis**
The preparation of the samples and their analysis were performed following the same procedures adopted for the 1992
Guliya core samples (Sierra-Hernández et al., 2018). Briefly, 159 ice samples comprising the years 1971 (11 m) to
2015 (0.06 m) were cut from the 309.73 m long-deep ice core. To ensure the analysis of 3–4 samples year$^{-1}$, the sample
resolution was adjusted between 4.5 and 11 cm accordingly. The samples were rinsed three times with nanopure water
(18.3 MΩ) in a class-100 cleanroom and placed in acid pre-cleaned LDPE containers (Nalgene) to melt. Once melted,
samples were transferred into acid pre-cleaned LDPE vials where nitric acid ($HNO_3$, Optima for Ultra Trace Element
Analysis, Fisher Scientific) was added to obtain a 2 % (v/v) acidified sample. The samples were then stored in the
cleanroom for a 30-day period during which the acid leaching process took place. At the end of the 30-day period, the
samples were immediately analyzed or stored at –32 °C.
Twenty-nine TEs (Ag, Al, As, Ba, Bi, Cd, Co, Cr, Cs, Cu, Fe, Ga, Li, Mg, Mn, Mo, Na, Nb, Ni, Pb, Rb, Sb, Sn, Sr,
Ti, Tl, U, V, and Zn) were measured in the samples by Inductively Coupled Plasma Sector Field Mass Spectrometry
(ICP-SFMS) (Element 2) (Sierra-Hernández et al., 2018; Beaudon et al., 2017). Trace elements were quantified using
linear calibration curves constructed from external standards analyzed before and after the samples.
Detection limits, procedural blanks, and accuracy results are presented in Table S1. Detection limits correspond to
three standard deviations of the concentration of 10 blank measurements (2 % optima $HNO_3$ aqueous solution) and
fluctuate between 0.01 pg g$^{-1}$ for Bi to 0.2 ng g$^{-1}$ for Fe and 0.4 ng g$^{-1}$ for Na (Sierra-Hernández et al., 2018; Beaudon
et al., 2017). To verify that the sampling and decontamination procedures did not add TEs to the ice core samples,
procedural blanks were made with nanopure water and analyzed with the ice core samples (Uglietti et al., 2014). Their
TE concentrations are considered negligible for all TEs, apart from Nb (9 %), as they were lower than 2 % of the
corresponding median concentration. The accuracy of the ICP-SFMS was determined each day of analysis using a 20-
fold dilution of a TMRain-95 certified solution (Environment Canada). The obtained TE concentrations fell within
the uncertainty limits in the certificate of analysis.



## 2.3 Non-crustal contribution


Enrichment Factor (EF) and Excess concentrations are used to assess the crustal and non-crustal (e.g., anthropogenic)
origins of each TE.
The EF is obtained following Eq. (1)
$$EF = [TE / Fe]_{ice} / [TE / Fe]_{PSA} \tag{1}$$
where $[TE / Fe]_{ice}$ corresponds to the ratio of a particular TE concentration to that of Fe in an ice sample and
$[TE / Fe]_{PSA}$ is the respective ratio in dust samples used as potential source area (PSA). For details about the Guliya
PSA and EFs derived from PSAs, the reader is referred to Sierra-Hernández (2018).
Similar to the previous Guliya TEs study, Fe was chosen as the crustal reference due to its stability and high abundance
in soil and rocks (Wedepohl, 1995), its high concentration both in the ice core samples and the PSAs; and the ability
of the ICP-SFMS to measure Fe with high accuracy and precision (Uglietti et al., 2014).
Excess concentrations are calculated following Eq. (2)
$$Excess = [TE]_{ice} - \left([TE/Fe]_{pre-industrial} \times [Fe]_{ice}\right) \tag{2}$$
$[TE]_{ice}$ and $[Fe]_{ice}$ are the concentrations of a particular TE and of Fe in the sample; $[TE/Fe]_{pre-industrial}$ is the median of
the TE concentration to the Fe concentration during the pre-industrial period (1650-1750), as obtained from the 1992
Guliya record (Sierra-Hernández et al., 2018).
To be consistent with our previous Guliya TEs publication, a TE will be considered of non-crustal origin (enriched)
when increases in EF and Excess concentration are significantly different from its background (pre-industrial levels),
using both a two-sample t-test for averages and the Mann–Whitney test for medians (p < 0.01).

## 2.4 Statistical analysis


All statistical analyses, factor analysis, cluster analysis, Mann–Whitney test for medians and a two-sample t-test for
averages and Mann–Kendall trend tests, were performed using Minitab 17 and 18. The Mann–Whitney test and the
two-sample t-test were applied to the entire data set subdivided into three groups: 1971–1990, 1990–2000, and 2000–
157 2015.

## 3 Results and discussion


The time series of Cd, Pb, Zn, Ni, and Al concentrations, Excess concentrations and EFs are presented as 5-year
running means in Fig. 2. The concentrations show high variability between 1971 and 1990 that decreases after 1990
perhaps as a result of the decreasing frequency of dust storms in the region (Thompson et al., 2018).





The Excess concentrations and EFs of Cd, Ni, Pb, and Zn increase after ~1990 and continue to increase more rapidly
and significantly after 2000. Their EF averages increase by ~10 % during 1990–2000, and during 2000–2015 by 75
% (Cd), 35 % (Pb), 30 % (Zn) and 10 % relative to the 1971–1990 period.
A comparison between the 1992 and the 2015 Guliya TE records is discussed in the Supplement (Fig. S1). The 1992
Guliya TE records show that enrichments of Pb and Cd begin ~1975 while the 2015 Guliya record shows they continue
to rise into the 21$^{st}$ century until ~2008 when the Cd enrichment started to decrease. In addition to these TEs, the 2015
record exhibits clear increases in Zn and Ni EFs since the 1990s into the 21$^{st}$ century, and similar to Cd, they decrease
after 2008. The Zn enrichment began to increase after 1975 similar to Pb and Cd; however, the signal may have been
overwhelmed by its crustal component in the 1992 core record rendering it undetectable.
A factor analysis method was used to assess the shared variability among TEs to determine possible common sources
(Sierra-Hernández et al., 2018). Much of the variance (94%) is explained by both Factor 1 (73 %) and Factor 2 (21
%) (Table S2). Similar to the 1992 TE results, TEs of crustal origin (e.g., Al, As, Ba, Fe, Mg, Mn, Ti, and V) fall into
Factor 1. In Fig. S2 the time series of Factor 1 scores are compared with the ice core concentrations of dust particles
($\rho = 0.20$, $p = 0.2$) and with the typical crustal TEs Fe and Al. Water-soluble TEs (e.g., Na, Sr), which are deposited
in the form of salts (evaporites) or carbonates, contribute to Factor 2 (Sierra-Hernández et al., 2018) (Table S2). This
is shown in the Factor 2 time series (Fig. S3) which have significant ($p < 0.001$) correlations with the ions Cl$^-$ ($\rho$ =
0.83), NO$_3^-$ ($\rho = 0.84$), SO$_4^{2-}$ ($\rho = 0.90$), Na$^+$ ($\rho = 0.92$), NH$_4^+$ ($\rho = 0.62$), K$^+$ ($\rho = 0.84$), Mg$^{2+}$ ($\rho = 0.86$), and Ca$^{2+}$ ($\rho$
= 0.75) (Thompson et al., 2018).
Factor 3 explains 2 % of the variance and is loaded in Cd, and to a lesser extent in Bi, Cu, Mn, Ni, Pb, Sn, Tl, and Zn.
Although 2 % represents a low variance possibly within the background noise, it could still have a physical
significance (Moore and Grinsted, 2009). In order to determine if Factor 3 is physically explainable, its time series
scores are plotted with the EFs of Cd, Pb and Zn in Fig. 3.
Factor 3 was found to be significantly ($p < 0.01$) correlated with the EFs of the following 12 metals: Cd ($\rho = 0.92$),
Zn ($\rho = 0.92$), Pb ($\rho = 0.80$), and Ni ($\rho = 0.80$) shown in Fig. 2, and Ag ($\rho = 0.62$), Bi ($\rho = 0.60$), Co ($\rho = 0.50$), Cr ($\rho$
= 0.60), Cu ($\rho = 0.51$), Mn ($\rho = 0.63$), Sn ($\rho = 0.74$), and Tl ($\rho = 0.64$). This indicates that Factor 3 explains the EFs
of the aforementioned metals. To distribute the TEs into associated groups, a cluster analysis was performed with
Factors 1-3 using the Ward linkage method and the Euclidan distance measure (Fig. S4). The cluster analysis shows
that Pb and Zn are strongly associated with Cd, Bi, and Mn suggesting these TEs likely have a non-crustal origin.
**3.1 21st Century anthropogenic sources**
The Mann–Kendall trend test was used to detect TEs with sustained and significant increasing trends in EF and Excess
concentration during the 1971–2015 period. The trend test indicated that Bi, Cd, Ni, Pb, Tl, and Zn, which are loaded
in Factor 3, have significant increasing EF trends but only Cd, Ni, Pb, and Zn have additionally significantly increasing
trends in Excess concentration. The EFs obtained here using PSAs are much smaller than those calculated using the
upper continental crust average (Wedepohl, 1995) and also smaller than those from ice cores with lower dust loads





compared to Guliya. Thus, it is necessary to determine which of the TEs mentioned above were significantly more
enriched during the 2000–2015 period. For this purpose, two different tests were used, the Mann–Whitney test and
the two-sample t-test ($p < 0.0005$). Both tests showed that the EFs and Excess concentrations for all four metals, Cd,
Zn, Pb, and Ni, are significantly higher during the 2000–2015 period than during the 1971–1990 period. Thus, the
following sections will specifically focus on these TEs and their possible sources.
Back-trajectory frequency distributions was determined to establish the origin of air parcels reaching the Guliya ice
cap. Back trajectories (7 days) were calculated daily for the 1992–2015 period for winter (December–January–
February) (Fig. 1b) and summer (June–July–August) (Fig. 1c) using the HYSPLIT model from the National Oceanic
and Atmospheric Administration. During winter, Guliya is strongly influenced by air parcels mostly from western
Xinjiang (China); from Central Asia, which consists of the former Soviet republics of Kazakhstan, Kyrgyzstan,
Tajikistan, Turkmenistan, and Uzbekistan; from Afghanistan and Pakistan in South Asia; and to a lesser extent from
the Middle East, Northern Africa, Eastern, and Western Europe. In summer, westerly and southerly (monsoonal)
flows, and even occasional northerly flows, influence Guliya, such that the entire Xinjiang region in addition to Central
Asia and the northern regions of Afghanistan and Pakistan lies within the back trajectories area. Air parcels from other
Southern Asian countries, such as India and Nepal, can also reach Guliya during summer.
Trace element enrichments in the Guliya core could reflect changes in emissions, atmospheric circulation, and/or post-
depositional processes. Post-depositional processes (e.g., seasonal surface melting, percolation and refreezing of
meltwater) do not significantly affect the stratigraphy of the Guliya core (Thompson et al., 2018). The Guliya borehole
temperatures were between –8°C and –12°C from the surface to ~15 m depth confirming that the ice is cold (Thompson
et al., 2018) and that overprinting of the TE records due to meltwater percolation is unlikely to occur. Thus, the
enrichments observed in the Guliya record indicate increasing emissions in specific source regions and/or changes in
atmospheric circulation.
To determine the origin of the Guliya Cd, Pb, Zn, and Ni enrichments, we examine the most important emission
sources of atmospheric TEs from the determined regions of influence: Central Asia, South Asia (Afghanistan,
Pakistan, and India), and Xinjiang (China). We also use $PM_{2.5}$ emission estimates, which very likely contain toxic
metals such as Cd, Pb, Ni, and Zn, using the EDGAR (Emissions Database for Global Atmospheric Research) v4.3.2
air pollutant dataset (1970–2012) (EDGARv4.3.2, 2017; Crippa et al., 2018). The EDGAR dataset provides total $PM_{2.5}$
and also $PM_{2.5}$ by emission sector for all countries.
In the EDGAR database, the total $PM_{2.5}$ corresponds to emissions from all human activities except large-scale biomass
burning and land use, land-use change, and forestry (EDGARv4.3.2, 2017; Crippa et al., 2018). To better understand
the possible emission sources, here we divided the EDGAR $PM_{2.5}$ emission sectors into four source categories in
accordance with the 2006 IPCC Guidelines for National Greenhouse Gas Inventories (IPCC, 2006). These source
categories include: 1) fossil fuel combustion that comprises the emission sectors: power generation and combustion
in the manufacturing, transportation, and residential sectors, 2) industrial processes which include the emission
sectors: mineral, chemical, and metal industry and other production industry (note: this category does not include any
type of fossil fuel combustion used by these industries), 3) agriculture which includes the emission sectors: manure





management, rice cultivation, direct soil emission, manure in pasture/range/paddock and other direct soil emissions,
and 4) waste that includes the emission sectors: waste incineration and solid waste disposal on land.
From 1980 to 2012, total $PM_{2.5}$ emissions in Afghanistan, Pakistan, India, Nepal, and China primarily originated from
the fossil fuel combustion emission sectors (~80–95 %), followed by the industrial source category (~5–20 %),
agriculture source category (~9 %) and waste source category (<1 % for China, no $PM_{2.5}$ emissions estimated for waste
incineration from the other countries in the EDGAR database) (EDGARv4.3.2, 2017; Crippa et al., 2018). Regarding
Xinjiang in particular, numerous studies estimated coal combustion as the major source of atmospheric Cd, Pb, Zn,
and Ni followed by smelting processes as a source for Cd, Pb, and Zn (Li et al., 2012; Shao et al., 2013; Cheng et al.,
2014; Tian et al., 2015), and by oil combustion as a source for Ni (Tian et al., 2012). Thus, in the following sections
we focus on the three largest TE and $PM_{2.5}$ source categories: fossil fuel combustion (Sect. 3.1.1), metal production
(Sect. 3.1.2), and agricultural sector (Sect. 3.1.3).

### 243 3.1.1 Fossil fuel combustion

In the regions that influence Guliya, two distinct trends in fossil fuel consumption and total $PM_{2.5}$ emissions are
discernible. Firstly, a steady increasing trend since the 1970s is observed in the Xinjiang province, Afghanistan,
Pakistan, India, and Nepal (Fig. 4a). Secondly, a decline after the 1990s is observed in Central Asian countries due to
the collapse of the Soviet Union (Fig. 4b). India and Xinjiang (China) became the largest consumers of both coal and
oil in the region during the 21st century, with coal as their primary energy source (Fig. 4). The third largest consumer
of coal and oil in the region during the 21st century is Kazakhstan despite its decreased consumption after the 1990s.
Interestingly, for each individual country, total $PM_{2.5}$ emissions generally follow coal consumption as shown in Fig.

251    4.

Significant positive correlations were found between the Guliya EFs of Cd, Pb, Zn, and Ni and coal consumption in
Xinjiang, India, China, and Pakistan; oil consumption in Xinjiang, India, China, Pakistan, and Turkmenistan; and total
$PM_{2.5}$ emissions from China, India, Pakistan and Afghanistan. These positive correlations were expected since these
records show generally increasing trends as shown in Fig. 4. Although Pakistan's fossil fuel consumption is 1–2 orders
of magnitude lower than that of Xinjiang and India, the Guliya TE enrichments closely resemble Pakistan's coal
consumption between 2005 and 2015. Both records peaked in 2008 after which they began to decrease suggesting
Pakistan's coal consumption could be one of the sources of anthropogenic TEs observed in the Guliya core.
To further investigate the role of fossil fuel combustion in Pakistan and in the other regions, we examined $PM_{2.5}$
emitted by the different sectors comprising fossil fuel combustion. The most interesting outcome is the temporal
correspondence between the Guliya TE enrichments and Pakistan's $PM_{2.5}$ emissions from fossil fuel combustion
associated with road transportation and manufacturing and construction (M-C) industries, which are the two largest
$PM_{2.5}$ emission sectors in Pakistan (Fig. 5). The enrichments of Cd, Pb, Zn, and Ni have two maxima, one in 2000
when $PM_{2.5}$ emissions from road transportation peaked, and the other in 2008 when $PM_{2.5}$ emissions from the M-C
industries peaked. After 2008, TE enrichments (except for Pb) and $PM_{2.5}$ emissions (road transportation and M-C
industries) decreased (Fig. 5). These temporal similarities suggest that the TE enrichments detected in the Guliya ice





core after 1995 may have primarily originated from the combustion of fossil fuels in Pakistan by road transportation
(since 1995) and by the M-C industries (since 2004).
Although Pakistan's oil and coal consumption is much lower than Xinjiang and India's, Pakistan is home to some of
the most polluted cities in the world (WHO, 2016) due to the lack of emission controls and air quality standards
(Colbeck et al., 2010; Sánchez-Triana et al., 2014). These cities include Peshawar, Rawalpindi, Lahore, Faisalabad,
and Pakistan's capital, Islamabad (Rasheed et al., 2014; WHO, 2016; Shi et al., 2018), all located in northern Pakistan
from which air parcels have been shown to strongly influence the Guliya site throughout the year. Air parcels from
Xinjiang and India, on the other hand, only reach Guliya during summer (Fig. 1c) which further suggests that Pakistan
is the likely dominant geographical origin of the Guliya TE enrichments.

### 276    3.1.2 Metal production

Like fossil fuel consumption, metal production has increased in Asia since the 1980s (Fig. S5) being China, India and
Kazakhstan the most important non-ferrous metal producers in the region and in the world (BGS, 2015). In China,
most of the non-ferrous metal production is located in the coastal regions while all Ni production is located in the
western region of China (Gansu, Xinjiang, Chongqing, Yunnan, and Liaoning provinces). Gansu, just east of Xinjiang,
produces 95 % of the total Ni production (Yanjia and Chandler, 2010). The Guliya TE enrichment trends do not
resemble those of metal production in China, Pakistan, India nor Kazakhstan (Fig. S5). Thus, although these important
metal production sources are relatively close to Guliya, they are likely not the primary source of the Guliya TE
enrichments.
$PM_{2.5}$ emissions by industrial processes contribute ~20 %, 10 %, and 5 % to the total $PM_{2.5}$ emissions in Pakistan,
China, and both India and Kazakhstan, respectively. Pakistan's $PM_{2.5}$ emissions by industrial processes peaked in
2008 similar to the Guliya TE enrichments, but they remained relatively stable after that while the Guliya TE
enrichments decreased (Fig. S5). Thus, while the increasing emissions from metal production could also influence the
TE depositions observed in Guliya, the metal production temporal trends and the industrial $PM_{2.5}$ emissions suggest
they are not the main sources of the Guliya TE enrichments.

### 291    3.1.3 Agricultural activities

Emissions from agricultural activities are an important source of atmospheric $PM_{2.5}$ worldwide (Lelieveld et al., 2015;
Bauer et al., 2016). Fertilizers and pesticides can be a direct (aerial spreading) or indirect (soil exposed to wind erosion)
source of toxic metals such as As, Cd, Cu, Cr, Pb, Ni, Zn, and others to the atmosphere (Nriagu, 1989b; Nriagu and
Pacyna, 1988). In particular, fertilizers derived from phosphate rocks contain heavy metal impurities such as Cd and
Pb that can contaminate agricultural soils (Mortvedt, 1995; Roberts, 2014). While consumption of phosphate fertilizers
decreased in Central Asia in the 1990s, it has been increasing in China, Pakistan, India, and Nepal since the 1970s
(Fig. S6). The Guliya TE enrichments do not resemble the phosphate fertilizer consumption records in these countries
nor their $PM_{2.5}$ from agricultural activities (Fig. S6). Thus, even though agricultural activities and their emissions have



also been increasing since the 1970s, their depositions at the Guliya ice cap are likely overwhelmed by those from
fossil fuel combustion in Pakistan.
**3.2 Atmospheric circulation**
In our previous Guliya TEs study (Sierra-Hernández et al., 2018), we observed a positive correlation between the
North Atlantic Oscillation (NAO) index and the EFs at Guliya suggesting non-crustal depositions originated from
regions to the west. Likewise, Thompson et al. (2018) determined possible positive linkages between NAO and the
2015 Guliya ice core temperature and snowfall proxies. Fig. 6 presents a comparison of the 1992 and 2015 Guliya EF
composites with the winter (DJF) NAO index and with coal production/consumption from Europe, and Pakistan
between 1800 and 2015. The extended Guliya EF composite shows two periods of enrichment: ~1850–1940 and 1970–
2015. We suggested that the former originated from coal consumption in Europe, which alongside the U.S. was a
major coal consumer at the time. The TE enrichment dropped to pre-industrial levels during 1940–1970 coinciding
with a negative NAO phase. This drop occurred at a time when atmospheric emissions in Europe reached a maximum
(Pacyna and Pacyna, 2001) suggesting that atmospheric circulation (NAO) had a stronger influence over Guliya than
the emission source(s) intensity. Post-1970s enrichments in both the 1992 and the 2015 Guliya records occur during
positive phases of NAO. Emissions control devices were introduced in Europe ~1970 decreasing their $PM_{2.5}$ emissions
(Pacyna et al., 2007; Pacyna and Pacyna, 2001) and consequently atmospheric emissions of TEs; however, TE
enrichments continue to increase in the Guliya ice core record. This divergence can be explained by changes in either
emission source intensity and/or atmospheric circulation, and indeed changes in both emission intensity and
atmospheric circulation have occurred post-1990s.
TEs enrichments start to decrease in 2009 while the NAO entered a slightly negative phase between 2010 and 2014.
Similarly, coal consumption has decreased in Pakistan since 2009, in Kazakhstan since 2012, and in Afghanistan since
2011 while in Xinjiang it continues to increase. In addition, the Chinese government implemented the 12[th] FYP (2011–
2015) that included policies to mitigate heavy metals emissions, implement air quality standards on $PM_{2.5}$, and increase
renewable energy consumption which likely explains the decrease of smoke and dust particles emitted in Xinjiang
since 2014 (Fig. 4). The decline in coal consumption in the region, the Chinese mitigation policies and the slightly
negative NAO could all have led to a decrease in EFs and Excess concentrations. However, the winter NAO index has
also been slightly negative in other years (e.g., 1919, 2000) without affecting the enrichments detected in the Guliya
core. Therefore, the slightly negative NAO phase between 2010 and 2014 probably did not play a significant role in
the Guliya TE enrichments decrease after 2009.
Although practically all human activities that emit $PM_{2.5}$ and toxic metals have been increasing in South Asia and in
Northwest China, it is very likely that emissions from Pakistan's fossil fuel combustion have dominated the Guliya
Cd, Pb, Zn, and Ni enrichments during the 21[st] century, and more specifically oil combustion by road transportation
since 1995 and the subsequent significant rise of coal combustion in 2004, likely mostly consumed by the M-C
industries. Emissions from other sources (industrial processes, agriculture activities, waste disposal) from Pakistan



and other regions (Central Asia, Xinjiang, Afghanistan, Northwest India and Nepal) have also likely been deposited
on the Guliya ice cap but to a lesser extent.
China is investing $62 billion in the Pakistan-China Economic Corridor (PCEP), established in 2013, to improve the
economy of Pakistan and to facilitate economic connectivity with other countries in the region. Until 2015, Pakistan's
energy was generated mostly from oil, gas, and hydropower. After the establishment of the PCEP, five coal-fired
power plants were built and have been operational since 2017. Three more are expected to open during 2019 and
others are currently under construction (CPEC, 2019). Moving to coal-generated power will further increase the total
coal consumption in Pakistan which in turn could have severe environmental and human health impacts in the region
if mitigation actions are not taken.

## 4 Conclusions

A new continuous, high-resolution ice core record of trace elements covering the 1971–2015 period was extracted
from the Guliya ice cap in Northwestern Tibet, China. This new record extends our previous 1650–1991 Guliya record
well into the 21$^{st}$ century making it, the first and most up to date ice core-derived archive of trace metal contamination
in the Third Pole region to date. Since the dust concentrations in the Guliya ice cores are extremely high in comparison
to other ice cores from the Third Pole as shown in Sierra-Hernandez et al. (2018), we also used EF and Excess
concentrations to differentiate between crustal and non-crustal origins in this new record. Increases in EF and Excess
concentrations of Cd, Pb, Zn, and Ni are observed since the 1990s reaching a maximum in 2008. The enrichments of
Cd, Pb, Zn, and Ni increased by ~75 %, 35 %, 30 %, and 10 %, respectively during the 2000–2015 period relative to
1971–1990. Comparisons between the Cd, Pb, Zn, and Ni enrichments from the Guliya records and fossil fuel
consumption, metal production, phosphate fertilizer consumption, and PM$_{2.5}$ emissions from Xinjiang (China),
Afghanistan, Pakistan, India, Nepal, and Central Asia suggest that the metal enrichments detected in Guliya originate
primarily from fossil fuel combustion (road transportation and the manufacturing and construction industry) in
Pakistan between 1995 and 2015. The post-2008 Cd, Zn, and Ni decline likely reflects a coal consumption decrease
in Pakistan at that time. It is likely that emissions from Xinjiang, Afghanistan, India, Nepal, and Central Asia, are also
impacting Guliya during the 21$^{st}$ century, however their contribution is overwhelmed by that of fossil fuel combustion
emissions from Pakistan. This new Guliya ice core record demonstrates that the current emissions in Asia are
impacting remote high-altitude glaciers in the region. Therefore, mitigation policies and technologies should be
enforced by the governments of Central and South Asian countries to improve the air quality in the region as most
Asian countries continue to develop.

**Data availability.**
**Supplement.** The Supplement is available online





**Author contributions.** R.S.H. wrote the paper, performed all the data analysis, interpreted the data, and ran the daily HYSPLIT back trajectories; R.S.H. and E.B. prepared the ice core samples and conducted their ICP-MS analysis; E.B. created the maps for Fig. 1. All authors contributed to the study design, data interpretation, revision and edition of the manuscript. P.G. oversees the ICP-MS lab. L.G.T. planned the drilling operation and led the field expedition in which both L.G.T. and P.G. contributed to the ice core processing in the field.

**Competing interests.** The authors declare no competing financial interests.

**Acknowledgments**

The NSF P2C2 program (1502919) funded this study. We thank everybody that made the 2015 Guliya field expedition a success, in particular Tandong Yao from the Institute for Tibetan Plateau Research. The authors greatly acknowledge Julien Nicolas for creating the seasonal back trajectory frequencies grid for all our Guliya TEs publications as well as for his time discussing them. We thank Aaron Wilson for his insights on the meteorology of the region. We thank Xiaoxing Yang and Chao You for providing recent energy yearbook data from Xinjiang. We also thank Max Woodworth for helpful discussions about the development and policies in Western China. We are grateful to Ellen Mosley-Thompson and Mary Davis for valuable comments throughout the development of the manuscript. Stacy Porter is greatly acknowledged for her help in editing and improving the English of the different versions of the manuscript. Lastly, we thank Henry Brecher for his detailed proofreading of the manuscript.

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


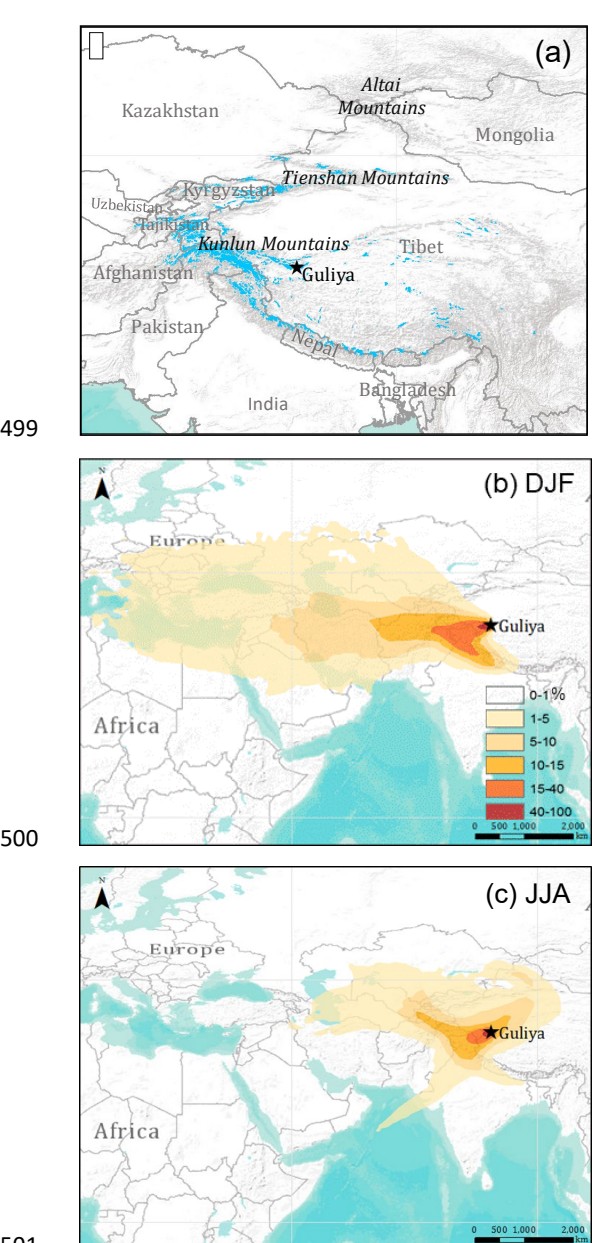




**Figure 1.** Maps showing the location of the Guliya ice cap (star) and the three mountain ranges of the Xinjiang
province (a), and seasonal NOAA HYSPLIT 7-day back trajectories frequency plots for December, January, and
February (b); and June, July, and August (c) for the 1992–2015 period (Rolph et al., 2017; Stein et al., 2015).





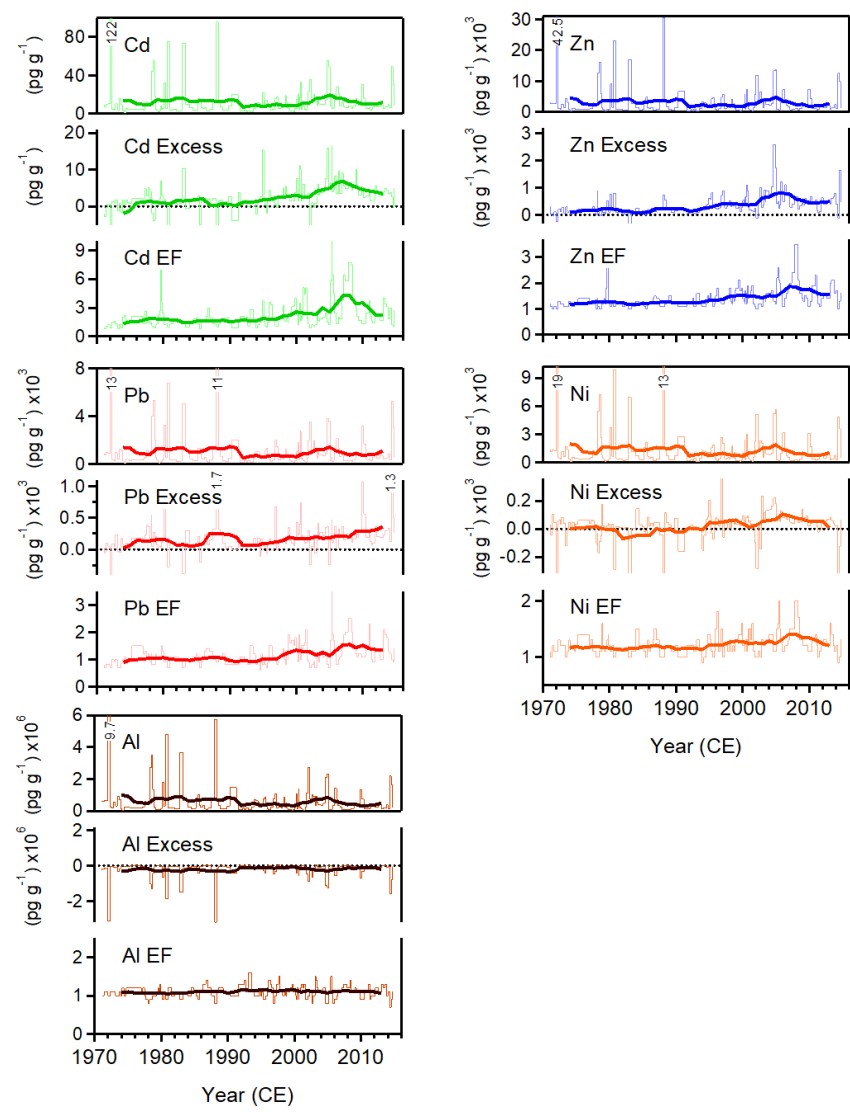

**Figure 2**. Cd, Pb, Zn, Ni, and Al 5-year running means (thick lines) of concentrations and EFs between 1971 and 2015. Thin lines show the sample resolution.



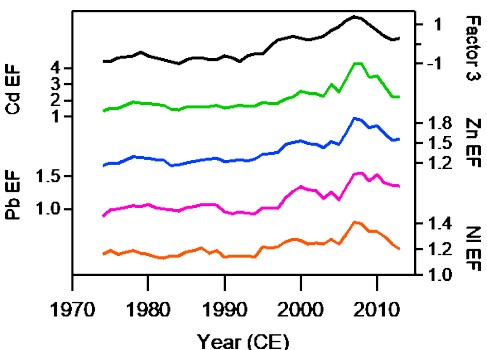


**Figure 3.** Comparison of 5-year running means of Factor 3 scores with Cd, Zn, Pb and Ni EFs between 1971 and

511    2015.









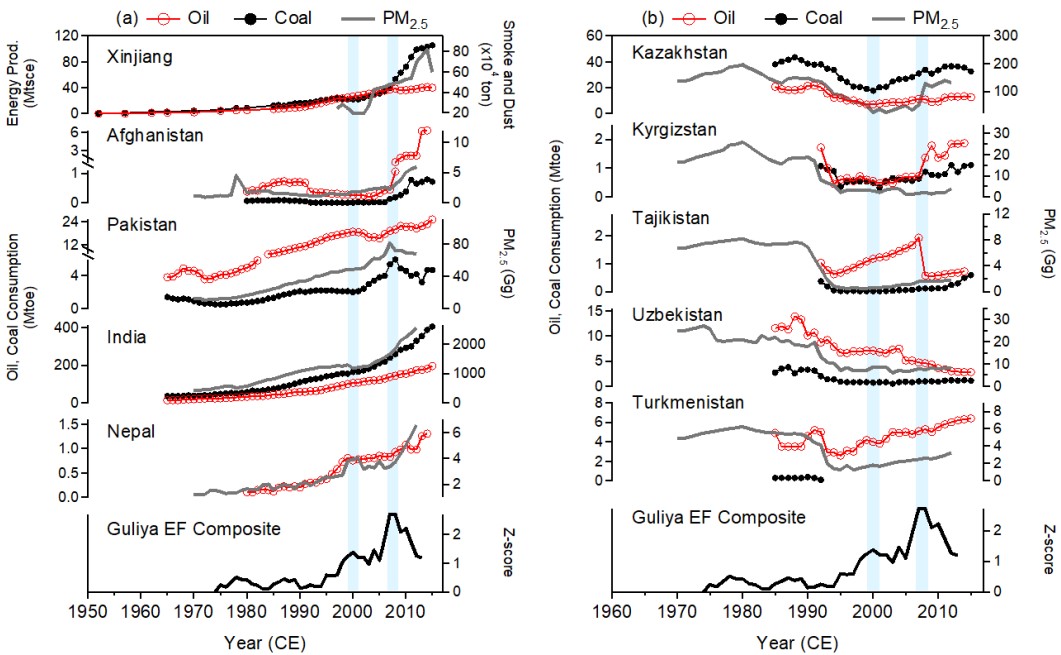


**Figure 4.** Oil and coal consumption in: (a) Xinjiang, as coal energy production in million tonnes of standard coal
equivalent (Jianxin, 2016); in million tonnes of oil equivalent (Mtoe) in Afghanistan (EIA, 2019), Pakistan (BP,
2016), India (BP, 2016), and Nepal (coal ≤ 0.1 Mtoe) (EIA, 2019). (b) Central Asian countries: Kazakhstan (BP,
2016), Kyrgizstan and Tajikistan (EIA, 2019), and Uzbekistan and Turkmenistan (BP, 2016). $PM_{2.5}$ emissions from
anthropogenic sources (EDGARv4.3.2, 2017; Crippa et al., 2018) shown in both panels for all countries except
Xinjiang (China). Smoke and dust emissions from Xinjiang (China) (Ning, 2016) are shown since no $PM_{2.5}$ data was
available. The Guliya EF composite (average of Cd, Pb, Zn, and Ni EF z-scores) is shown at the bottom of each
panel for comparison. The two Guliya maxima at 2000 and 2008 are shown as shaded bars.






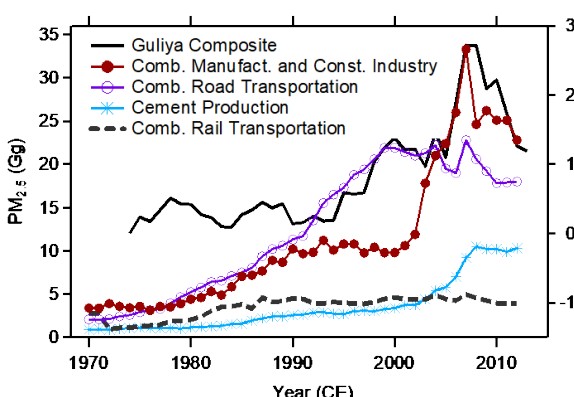


**Figure 5.** Larges emitter sectors of PM$_{2.5}$ in Pakistan between 1970 and 2012: fossil fuel combustion in

manufacturing and construction industries, fossil fuel combustion by road transportation, cement production, and

fossil fuel combustion by rail transportation (EDGARv4.3.2, 2017; Crippa et al., 2018). The Guliya EF composite

(average of Cd, Pb, Zn, and Ni EF z-scores) is shown for comparison.













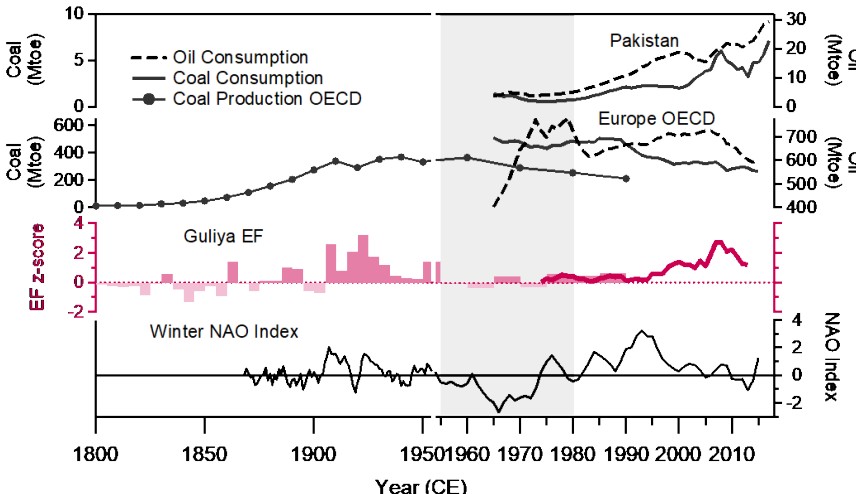

**Figure 6.** Comparisons of the Guliya EF composite (average of Cd, Pb, Zn, and Ni EF z-scores) from the 2015
Guliya core with the winter NAO index shown as 5-year running mean (Hurrell, 2003). The 1992 Guliya EF
composite (average of Cd, Pb, and Zn EF z-scores) is shown as 5-year medians and the 2015 Guliya EF composite is
a 5-year running mean. The light gray band (1955–1980) highlights the period of negative NAO. Pakistan's and
Europe coal and oil consumption (1965–2017) (BP, 2016) are shown for comparison. Europe's coal production
(1800–1990) (HYDE, 2006) is also shown. Mtoe is million tonnes of oil equivalent.
