# Peer review of "21st Century Asian air pollution impacts glacier in northwestern"

_Atmospheric Chemistry and Physics, 2019_

## Referee Comment (RC1) · Aubrey Hillman (Referee) · 18 Jul 2019

Overall I think Sierra-Hernandez and others present a comprehensive and interesting follow-up of the 1992 Guilya ice core and highlight some important trends in trace metals that have taken place since then. In general I only have minor comments and suggestions. The largest remaining question in my mind is how an NAO index is mechanistically related to higher EFs, so an explanation of that would be helpful.

Line 56- What is the "new Silk Road"? Line 65- Is this percentage (50-60%) meant to imply this is how much emissions can be attributed to motor vehicles? Lines 93-95- If more anthropogenic sources have emerged, then what new developments have taken place and are in use in this manuscript to attribute possible sources? Lines

139-140- Fe is an interesting choice. Why not something else like Al? Or Ti? Lines 150-152- What is defined as "pre-industrial" in this case? Before what time period? Lines 192-194- I understand the difference between EF and Excess calculations, but why would some elements have significant trends for EF but not Excess? This needs some additional explanation. Lines 224-225- Since the EDGAR database excludes biomass burning and land use change, is it possible that some of the observed trends could be attributed to these processes? I know it will be hard to make these estimations without quantitative data, but these could be potentially significant. Lines 303-305- A brief mechanistic explanation about how a positive NAO index actually results in higher EFs would be useful. Lines 310-313- However, I think it's important to note that the drop from 1940-1970 is also when there is a gap/transition in the data. Neither coal nor oil consumption estimates extend fully back through this period. And while coal production is still reasonably high, it does start to decline. This is not to say that I don't believe the NAO is having an impact, but I think it's important to highlight that there are some gaps in the data. Lines 337-336- This is an interesting development and would seem to be a reversal of coal consumption declines since 2009 as previously noted in line 320. Was the decline in 2009 a temporary slow down then? Figure 5- The second y-axis on the right needs a label. Figure 6- Why is the 1992 core plotted as a 5-year running median while the 2015 core is a 5-year running mean?

---

## Referee Comment (RC2) · Anonymous Referee #2 · 6 Aug 2019

Checklist of ACP review criteria:

>Does the paper address relevant scientific questions within the scope of ACP?

Yes.

>Does the paper present novel concepts, ideas, tools, or data?

New data yes, otherwise, not really. The overall approach, methods, and interpretation of the findings, including the manner in which the results are graphically presented, are very similar to what was reported in the 2018 paper by the same authors on the older (1992) Gulyia ice core. It is understandable that the methods need to be consistent to ensure a coherent continuity between the older core and the new extension. However I can't help to wonder why the authors did not simply report findings from the two

cores together in a single paper. There is a great deal of unnecessary duplication of information in this new paper that is not really novel, but a simple repetition of earlier work.

>Are substantial conclusions reached?

Somewhat. This is not the first ice core record of trace element deposition to have been developed from central or East Asia (cf Fig. 4 in Sierra-Hernandez et al., 2018), and it shows a continuation of the same general trend in increasing trace metal deposition in the late 20th century seen in other cores, which generally agrees with what is know of regional emission trends from possible anthropogenic sources. However, the analysis of this set of cores from Gulyia ice cap was, in my view, done with exceptional care and attention to data quality compared to previous studies.

>Are the scientific methods and assumptions valid and clearly outlined?

Yes, except in the discussion of the potential role of the North Atlantic Oscillation (NAO) on the atmospheric transport of trace elements (section 3.2). This part of the paper is weak and unconvincing, and lacks clarity. See specific comments below.

>Are the results sufficient to support the interpretations and conclusions?

Yes, but with the same proviso as above regarding the NAO.

>Is the description of experiments and calculations sufficiently complete and precise to allow their >reproduction by fellow scientists (traceability of results)?

Overall, yes, or adequate references to earlier publications with details are provided. However I have some doubts about the method of calculation of the "Excess" of trace elements in the ice core. See specific comments below.

>Do the authors give proper credit to related work and clearly indicate their own new/original contribution?

Yes.
>Does the title clearly reflect the contents of the paper?

Yes.

>Does the abstract provide a concise and complete summary?

Yes.

>Is the overall presentation well structured and clear?

Yes, but the introduction section repeats much of the same information that was previously given in Sierra-Hernandez et al. (2018), and seems unnecessarily long and wordy to me.

>Is the language fluent and precise?

Yes.

>Are mathematical formulae, symbols, abbreviations, and units correctly defined and used?

Yes, but I have some questions about the calculation of "excesses" of TEs, see specific comments below.

>Should any parts of the paper (text, formulae, figures, tables) be clarified, reduced, combined, or >eliminated?

I recommend shortening the introduction, abbreviating the discussion in section 3.1, and dropping section 3.2 altogether. See specific comments below.

>Are the number and quality of references appropriate?

Yes.

>Is the amount and quality of supplementary material appropriate?

Yes.
Specific comments:

I reviewed an earlier version of this paper which had been submitted for consideration in another journal. I had made a number of recommendations for improving the paper, many of which were implemented by the authors in this new version. I focus below on some of the recommendations that were not followed, and on other points.

In my previous review, I recommended that since the enrichment factors (EFs) were calculated using Fe as a reference element, the authors should show that Fe concentrations in the overlapping sections of the 2015 and 1992 Gulyia ice cores (i.e., for 1971-1992) are comparable. In the revised version of the paper, the authors indicate (in the Supplement) that "the AI and Fe median concentrations are 0.3  $\mu$ g/g in both records during the 1971–1991 period in which both TE records overlap." However the data are not actually shown. I recommend again that these data should be presented graphically. This would show the reader if the Fe concentrations in the overlapping core sections vary in agreement. The fact that they have the same median concentrations does not necessarily imply that they do.

L187-189: Results of cluster analyses of geochemical data should be treated with a great deal of caution, as they are very sensitive to data pre-treatment (example: transformations) and the choice of the clustering algorithm. See Templ et al. (2008; Applied Geochemistry 23: 2198–2213). The results presented on Fig. S4 would be more convincing if the authors could show that they can be replicated with different clustering criteria or methods.

On Fig. S1, aluminum (AI) shows consistently negative Excesses (i.e., deficits) in the older Gulyia (1992) ice core, including during the "pre-industrial" interval of reference (1650-1750). This is odd. The concentration of trace metals of crustal origin in environmental matrices (e.g., soils, ice) often have positively skewed probability distributions, but not that of major elements such as AI and Fe. Therefore I would expect the probability distribution of the AI/Fe ratio in the ice to be symmetrical, possibly normal. Hence,
if the Ex(AI) shown on Fig. S1 were calculated as per equation (2) in the paper, I would also expect that in the part of the core that corresponds to the interval of reference (1650-1750) there should be both positive and negative Ex(AI), depending on whether the AI/Fe ratio measured in any part of the core fell above or below the median AI/Fe value for the whole reference interval. In order for the calculated AI Excess to be consistently negative during this interval of reference, the measured AI/Fe ratios must be consistently lower than the reference median AI/Fe value used in the calculation, which necessarily implies that this value can't actually be the median. Something seems to be wrong here, and it may also affect the calculation of Excesses for other trace elements. Maybe there is something missing in the description of the calculation method ?

Sections 3.1 and Figs. 4-5. The attribution of anthropogenically-derived trace metals deposited in the Gulyia ice core to specific regional sources is largely based on visual "curve matching" of trends between the ice-core composite EF (for Cd, Pb, Zn, and Ni) and in regional emission data. This is fair enough, and I think that there is a good case to be made that the increase in the composite EFs points to dominant sources from East Asia and/or the Indian sub-continent (which is hardly surprising). I am less convinced by the argument offered for the predominance of emissions from specific fossil fuel sources in Pakistan. The observed trend in EFs is probably the result of a mixture of emissions from various regions/sectors. Hence, there could in fact be more than one combination of regional/sector emission sources that could produce the observed trend in EF, but the only such combination that is analyzed in detail is that of emissions from Pakistan (Fig. 5). The argument offered in support of Pakistan would be more convincing if it could be shown that no other combination of regional sources can explain the observed trend in EFs. Ultimately, the "case" for the predominance of Pakistan seems to depend on the apparent "peak" in EFs around 2007, which could match a period of peak emissions in Pakistan at that time (if emission data from this region are to be trusted). Given that the factor analysis attributes only 2 % of the variance in TEs to anthropogenic sources (the rest being associated with crustal sources), the

**ACPD**
authors should refrain from over-interpreting minor features in the TE record. This is not to say that Pakistan may not be an important source of TEs to the Gulyia ice cap (it would be surprising if it were not), but I think that the relative dominance of emissions from this region is overstated.

Section 3.2.: I find the discussion of the possible influence of the NAO on atmospheric trace element deposition on Gulyia ice cap to be weak. First, the purported correspondence between high/low NAO phases and the composite index of trace metal EF on the Gulyia ice cap (Fig. 6) is based on a subjective visual comparison, without any supporting qualitative metrics (e.g., correlation coefficients), and it is, to me, unconvincing. Second, this comparison does not offer a definitive way to discriminate or parse, in a quantitative way, the relative influences of the NAO and of anthropogenic source emissions of trace elements, such that one is left to speculate about which factor(s) were dominant at different times. Thirdly, no explicit mechanism is offered in the text to account for this purported relationship. I am assuming the interpretation is the same as that previously suggested in Sierra-Hernandez et al. (2018), i.e. stronger wintertime NAO => enhanced westerlies => more efficient transport of atmospheric trace elements from distant (European) sources in the west to the Gulyia ice cap. As I pointed out in my previous review, this is at odds with the effects of the NAO on atmospheric flow over the Tibetan Plateau in climatology publications (Mao et al., 2011; doi:10.1016/j.atmosenv.2010.10.020; Han et al. 2008; doi:10.1016/j.atmosenv.2007.12.025). Furthermore, there seems to be no clear or consistent association between predominantly low(high) NAO phases and variations in dust deposition on the Gulyia ice cap, as one might expect if the NAO-westerlies linkage was important for atmospheric particulate matter transport (Fig. 6-7 in Thompson et al., 2018). I had previously suggested that one possible way to verify if a stronger winter/spring NAO phase actually enhances east-west atmospheric transport towards the ice cap would be to compare the mean length of air parcel back-trajectories between years of low and high NAO indices. I can only offer the same suggestion again. Unless this or some other supporting evidence can be offered, I recommend that this

**ACPD**
section be excised altogether from the paper.

Minor suggested corrections:

L32-34: "TEs are also released into the atmosphere by human activities including: 1) the combustion of fossil fuels including coal, oil and its distillates (e.g., gasoline, jet fuel, diesel); 2) industrial processes such as mineral EXTRACTION, and metal production."

L163-164: "Their EF averages increase by  ${\sim}10$  % during 1990–2000, and during 2000–2015 by 75 % (Cd), 35 % (Pb), 30 % (Zn) and 10 % relative to the 1971–1990 period."

What does the 10 % figure refer to ? Ni ?

L172: "MOST of the variance (94%) is explained by both Factor 1 (73 %)...."

---

## Referee Comment (RC3) · Anonymous Referee #3 · 10 Aug 2019

The work of Sierra-Hernández et al. describes trace metal analysis of an ice core from the Guliya ice cap in northwestern Tibet that represents deposition from 1971 to 2015. This extends the metals deposition record from that location that previously ended at 1991. The authors demonstrate increased deposition of several trace metals in the 21st century, with some starting to decrease more recently. Trends in deposition are related to industrial activities in the region and to climate. The measurements are made with good analytical rigor, but some of the trends and source attribution could be more convincing. I recommend the changes detailed below be made to the manuscript prior to consideration for publication in ACP.

General comments

The Authors use two different metrics to assess enrichments of metals, enrichment

factor (EF) and excess concentration. The rationale for using two metrics (i.e. the unique information provided by each) is not provided. It would be useful to provide additional information of this type in Section 2.3. Where different trends were observed for each metric (e.g. lines 191-194, Figure 2), the authors should explain the reasoning and implications of these differences.

Section 3.2 relating the North Atlantic Oscillation (NAO) to enrichments on the Guliya ice cap is very qualitative and speculative. Although it seems reasonable that some relationship exists with the NAO, there is no quantitative analysis provided. The authors mention that correlation was observed in previous studies (lines 303-305). Was there any correlation observed here? It seems a stretch to conclude that the source of enrichments for several trace metals is related to a specific industry in Pakistan (i.e. line 330).

Specific comments

Line 46: Suggest not using an acronym for five-year plan since the term is used only three times in the manuscript.

Lines 55-59: Although the sampling site is given with reference to Xinjiang Province, this area is not labeled on the map in Figure 1(a). This should be added.

Lines 125-133: The quality assurance/quality control of the sample analysis is comprehensive and well-described, demonstrating the quality of the presented data.

Lines 140-141: Please provide a brief description of the PSA used here along with the reference to the previous study.

Lines 165-167: What about Pb deposition post-2008? The first part of the sentence refers to both Pb and Cd, while the final part of the sentence describes only Cd. Please clarify.

Line 252: Statistical information for the significant positive correlations should be provided.

Line 530: Typo in first word of the caption.

---

## Author Comment (AC1) · 11 Oct 2019

**Aubrey Hillman (Referee)**

**aubrey.hillman@louisiana.edu**

Received and published: 18 July 2019

Overall I think Sierra-Hernandez and others present a comprehensive and interesting follow-up of the 1992 Guilya ice core and highlight some important trends in trace metals that have taken place since then. In general I only have minor comments and suggestions. The largest remaining question in my mind is how an NAO index is mechanistically related to higher EFs, so an explanation of that would be helpful.

**Line 56- What is the "new Silk Road"?**

[Response] It is an initiative proposed by President Xi Jinping formally called the Belt and Road Initiative to modernize and build new railways, ports, pipelines, power grids and highways to connect China with the west. This has been added accordingly.

**Line 65- Is this percentage (50-60%) meant to imply this is how much emissions can be attributed to motor vehicles?**

[Response] Yes, we have changed it to "(estimated to be  $\sim$ 60–70 % of the total air emissions)". There was also a typo and it should be 60-70 % and not 50-60 %.

**Lines 93-95- If more anthropogenic sources have emerged, then what new developments have taken place and are in use in this manuscript to attribute possible sources?**

[Response] New comprehensive emission inventories of air pollutants have been developed since the 1970s and 1980s to regulate their emissions. In particular, we use the Emissions Database for Global Atmospheric Research (EDGAR v4.3.2) which compiled a comprehensive dataset of air pollutants between 1970 and 2012 and is used in this manuscript to attribute possible sources along with emission inventories from the U.S. Energy Information Administration, and the British Petroleoum (BP) Statistical Review of World Energy. In our previous publication (Sierra-Hernandez, 2018), the TE ice core record stopped at 1991, and thus the increases observed between 1975 and 1991 were not possible to compare with emission inventories since the time interval was too short. With this new Guliya core drilled in 2015, we now have a 40-year timeseries of TEs that can better show temporal trend similarities with emission inventories. A description of these new developments is now included in the text.

**Lines 139-140- Fe is an interesting choice. Why not something else like Al? Or Ti?**

[Response] In our previous publication (Sierra-Hernandez, 2018) we used Fe as a crustal element. To be consistent, we continue to use Fe as a crustal element in this new ice core record. In lines 143-145 we explain that "Fe was chosen as the crustal reference due to its stability and high abundance in soil and rocks (Wedepohl, 1995), its high concentration both in the ice core samples and the PSAs, and the ability of the ICP-SFMS to measure Fe with high accuracy and precision". We added the following text in lines 149-152: "Additionally, Fe is highly correlated with Al (r = 1), and also with Ba (r = 0.98). Like in Sierra-Hernandez (Sierra-Hernández et al., 2018), EFs calculated using Al and Ba as crustal references had no significant differences compared to EFs relative to Fe which all together shows that the choice of Fe as a crustal TE to calculate EFs did not affect the results."

Here is the comparison between EF relative to Fe (color curves) with EFs relative to Al (black curves)

**Lines 150-152- What is defined as "pre-industrial" in this case? Before what time period?**

[Response] We consider the pre-industrial period to be the period before the Industrial Revolution in Europe (~1780). Similar to our previous publication (Sierra-Hernandez, 2018), we used the period 1650-1750 as pre-industrial to be more conservative due to the time scale uncertainty of the ice core. We have clarified this in the text accordingly.

Lines 192-194- I understand the difference between EF and Excess calculations, but why would some elements have significant trends for EF but not Excess? This needs some additional explanation. [Response] For all our statistical tests we used the datasets at full resolution. However, the Mann-Kendall trend tests were performed with the annual dataset. The problem with the Excess concentration was that by averaging the values within one year, large negative values skewed the final results to negative numbers. The tests have been re-done using the full resolution EF and Excess concentration datasets. The updated results show that both EFs and Excess concentrations of Bi, Cd, Ni, Pb, Tl, and Zn show significant increasing trends. Changes were made in the manuscript accordingly.

**Lines 224-225- Since the EDGAR database excludes biomass burning and land use change, is it possible that some of the observed trends could be attributed to these processes? I know it will be hard to make these estimations without quantitative data, but these could be potentially significant.**

[Response] The EDGAR database does include some anthropogenic biomass burning activities such as domestic combustion and agricultural waste burning excluding only large-scale biomass burning. We have added a Biomass Burning section (Sect. 3.1.2) to discuss domestic combustion and agricultural waste burning.

It is possible that the TE trends observed in the Guliya ice core can be attributed to large-scale biomass burning and land use change. Recent increases in wildfires in the Himalayas have been detected in Central Tibet possibly due to recent warming (You et al., 2018). This is mentioned in Sect. 3.1.2.

**Lines 303-305- A brief mechanistic explanation about how a positive NAO index actually results in higher EFs would be useful.**

[Response] We have eliminated Section 3.2. Please see the full response regarding this in the responses to Reviewer #2.

Lines 310-313- However, I think it's important to note that the drop from 1940-1970 is also when there is a gap/transition in the data. Neither coal nor oil consumption estimates extend fully back

**through this period. And while coal production is still reasonably high, it does start to decline. This is not to say that I don't believe the NAO is having an impact, but I think it's important to highlight that there are some gaps in the data.**

[Response] Yes, we agree that the gaps in the inventories are important and should be noted. As described above the NAO discussion has been eliminated, therefore these inventories are not shown/discussed anymore.

**Lines 337-336- This is an interesting development and would seem to be a reversal of coal consumption declines since 2009 as previously noted in line 320. Was the decline in 2009 a temporary slow down then?**

[Response] Yes, it is a temporary slow-down between 2009 and 2013. This Figure has been eliminated.

**Figure 5- The second y-axis on the right needs a label.**

[Response] Label has been added. The Figure was moved to the Supplement as Fig. S6

**Figure 6- Why is the 1992 core plotted as a 5-year running median while the 2015 core is a 5-year running mean?**

[Response] The 2015 Guliya EF composite z-scores were plotted as 5-year running means in Fig. 6 to keep consistency with the previous figures of the manuscript. Since Section 3.2 has been removed, there is no Figure 6 anymore.

References

You, C., Yao, T., and Xu, C.: Recent Increases in Wildfires in the Himalayas and Surrounding Regions Detected in Central Tibetan Ice Core Records, J. Geophys. Res.: Atmospheres, 123, 3285-3291, 10.1002/2017jd027929, 2018.

---

## Author Comment (AC3) · 11 Oct 2019

**The work of Sierra-Hernández et al. describes trace metal analysis of an ice core from the Guliya ice cap in northwestern Tibet that represents deposition from 1971 to 2015. This extends the metals deposition record from that location that previously ended at 1991. The authors demonstrate increased deposition of several trace metals in the 21st century, with some starting to decrease more recently. Trends in deposition are related to industrial activities in the region and to climate. The measurements are made with good analytical rigor, but some of the trends and source attribution could be more convincing. I recommend the changes detailed below be made to the manuscript prior to consideration for publication in ACP.**
[Response] Thank you. Following the recommendations by this and the other reviewers we changed the source attribution. Please see our responses below.

**General comments**
**The Authors use two different metrics to assess enrichments of metals, enrichment factor (EF) and excess concentration. The rationale for using two metrics (i.e. the unique information provided by each) is not provided. It would be useful to provide additional information of this type in Section 2.3. Where different trends were observed for each metric (e.g. lines 191-194, Figure 2), the authors should explain the reasoning and implications of these differences.**
[Response] The EFs calculated relative to the PSA are particularly small since the composition of the PSA is a much closer representation of the crustal background of the ice samples compared to those obtained using the UCC (upper continental background by Wedephol). Thus, to further demonstrate that these "small" increases are significant, we used a second metric which is the Excess concentration. The excess concentration provides the TE concentration difference between TE deposition after and before the pre-industrial period. For the pre-industrial period we use the 1992 Guliya ice core data comprised between 1650 and 1750. We added a brief explanation in Section 2.3 accordingly.

The difference in trend observed between EF and Excess concentration was due to an error in the Mann-Kendall trend test. Here it is the response to Reviewer #1 to this issue:
For all our statistical tests we used the datasets at full resolution. However, the Mann-Kendall trend tests were performed with the annual dataset. The tests have been re-done using the full resolution EF and Excess concentration datasets. The updated results show that both EFs and Excess concentrations of Bi, Cd, Ni, Pb, Tl, and Zn show significant increasing trends.
The problem with the Excess concentration was that by averaging the values within one year, a large negative value skewed the final results to negative numbers.

**Section 3.2 relating the North Atlantic Oscillation (NAO) to enrichments on the Guliya ice cap is very qualitative and speculative. Although it seems reasonable that some relationship exists with the NAO, there is no quantitative analysis provided. The authors mention that correlation was observed in previous studies (lines 303-305). Was there any correlation observed here? It seems a stretch to conclude that the source of enrichments for several trace metals is related to a specific industry in Pakistan (i.e. line 330).**
[Response] We have eliminated Section 3.2. Please see the full response regarding this in the responses to Reviewer #2.
We agree with this reviewer (and reviewer #2) that the EF increasing trend might be a combination of regions and sectors, so we have changed throughout the discussion, abstract and conclusions the attribution of TEs to all the regions that influence Guliya (South Asia, western China, and Central Asia).

**Specific comments**

**Line 46: Suggest not using an acronym for five-year plan since the term is used only three times in the manuscript.**
[Response] Done.

**Lines 55-59: Although the sampling site is given with reference to Xinjiang Province, this area is not labeled on the map in Figure 1(a). This should be added.**
[Response] Done.

**Lines 125-133: The quality assurance/quality control of the sample analysis is comprehensive and well-described, demonstrating the quality of the presented data.**
[Response] Thanks for the comments.

**Lines 140-141: Please provide a brief description of the PSA used here along with the reference to the previous study**.
[Response] A description has been added accordingly.

**Lines 165-167: What about Pb deposition post-2008? The first part of the sentence refers to both Pb and Cd, while the final part of the sentence describes only Cd. Please clarify.**
[Response] The final part of the sentence has been changed and it now reads "The 1992 Guliya TE records show that enrichments of Pb and Cd begin ~1975 while the 2015 Guliya record shows they continue to rise into the 21st century until ~2008 when they started to decrease."

**Line 252: Statistical information for the significant positive correlations should be provided.**
[Response] Done. The significance is $p < 0.001$, and it has been added in the corresponding line.

**Line 530: Typo in first word of the caption.**
[Response] Done.

---

## Author Comment (AC2)

Anonymous Referee #2 Received and published: 6 August 2019

Checklist of ACP review criteria:

>Does the paper address relevant scientific questions within the scope of ACP? Yes.

**>Does the paper present novel concepts, ideas, tools, or data?**

New data yes, otherwise, not really. The overall approach, methods, and interpretation of the findings, including the manner in which the results are graphically presented, are very similar to what was reported in the 2018 paper by the same authors on the older (1992) Gulyia ice core. It is understandable that the methods need to be consistent to ensure a coherent continuity between the older core and the new extension. However I can't help to wonder why the authors did not simply report findings from the two C1 cores together in a single paper. There is a great deal of unnecessary duplication of information in this new paper that is not really novel, but a simple repetition of earlier work.

[Response] The work on trace elements of the 1992 Guliya ice core was done as part of a project called "Impact of Atmospheric Trace Elements on the "Third Pole" Glaciers". When this project was developed and approved by the NSF back in 2012, we did not have the 2015 Guliya ice core. In fact, we did not have the 2015 ice core by the time we had finished the analysis. And it was because of the results of the 1992 Guliya core and the publication itself showing a small increase in Cd and Pb enrichments after 1970 that we thought the 2015 Guliya core could help us to understand those enrichments.

**>Are substantial conclusions reached?**

Somewhat. This is not the first ice core record of trace element deposition to have been developed from central or East Asia (cf Fig. 4 in Sierra-Hernandez et al., 2018), and it shows a continuation of the same general trend in increasing trace metal deposition in the late 20th century seen in other cores, which generally agrees with what is know of regional emission trends from possible anthropogenic sources. However, the analysis of this set of cores from Gulyia ice cap was, in my view, done with exceptional care and attention to data quality compared to previous studies.

>Are the scientific methods and assumptions valid and clearly outlined?

Yes, except in the discussion of the potential role of the North Atlantic Oscillation (NAO) on the atmospheric transport of trace elements (section 3.2). This part of the paper is weak and unconvincing, and lacks clarity. See specific comments below. [Response] We have addressed the NAO comments below.

>Are the results sufficient to support the interpretations and conclusions? Yes, but with the same proviso as above regarding the NAO.

>Is the description of experiments and calculations sufficiently complete and precise to allow their reproduction by fellow scientists (traceability of results)?

Overall, yes, or adequate references to earlier publications with details are provided. However I have some doubts about the method of calculation of the "Excess" of trace elements in the ice core. See specific comments below.

[Response] We addressed the issue with the Excess concentration. Thank you for bringing this to our attention.

>Do the authors give proper credit to related work and clearly indicate their own new/original contribution?

Yes.

>Does the title clearly reflect the contents of the paper? Yes.

>Does the abstract provide a concise and complete summary? Yes.

>Is the overall presentation well structured and clear?

Yes, but the introduction section repeats much of the same information that was previously given in Sierra-Hernandez et al. (2018), and seems unnecessarily long and wordy to me. [Response] Please see below.

>Is the language fluent and precise? Yes.

>Are mathematical formulae, symbols, abbreviations, and units correctly defined and used? Yes, but I have some questions about the calculation of "excesses" of TEs, see specific comments below.

[Response] We address this point below.

>Should any parts of the paper (text, formulae, figures, tables) be clarified, reduced, combined, or eliminated?

I recommend shortening the introduction, abbreviating the discussion in section 3.1, and dropping section 3.2 altogether. See specific comments below.

[Response] We think that while the introduction might indeed be a bit long, it is necessary to put into context the importance of trace elements, the changes in the region that could lead to increases in atmospheric toxic metals, and what we did and found in the 1992 Guliya core. We acknowledge that some of the information was already given in our previous publication, however the reader of this new manuscript might not necessarily know it and might not necessarily go back to it. We did shorten it and make it less wordy as the reviewer said. We eliminated the following paragraph

"The rapid economic growth of China has been the result of its economic reform and open policy beginning in 1978 after Mao Zedong's death and the subsequent five-year plans (FYP) implemented by the Chinese government. With their economic growth, China and India started to increase their coal consumption since ~1970 amplifying the global atmospheric emissions of  $CO_2$  and  $PM_{2.5}$ "

**And the following lines:**

"The number of registered vehicles increased by ~400 % between 1996 (3.838 million) and 2014 (15.168 million) (Bajwa, 2015). Emissions from motor vehicles (60–70 %), industry, and the generation of ~54,000 t of solid waste per day which is either dumped or incinerated have been estimated as the principal sources of  $PM_{2.5}$  and air pollution in Pakistan"

Regarding Section 3.1, we also feel that is important to discuss the similarities with the emissions from Pakistan for instance (Figure 5). While we cannot conclude that Pakistan is the dominant source, as suggested by the reviewers, it shows the reader that it is a possibility and perhaps they might look further into this. We mentioned that we also investigated the emission sectors from other countries. However, we do not show all of them as there is no similarity.

>Are the number and quality of references appropriate? Yes.

>Is the amount and quality of supplementary material appropriate? Yes.

**Specific comments:**

I reviewed an earlier version of this paper which had been submitted for consideration in another journal. I had made a number of recommendations for improving the paper, many of which were implemented by the authors in this new version. I focus below on some of the recommendations that were not followed, and on other points.

[Response] Thank you for reviewing the manuscript once again and for useful comments and suggestions both times.

In my previous review, I recommended that since the enrichment factors (EFs) were calculated using Fe as a reference element, the authors should show that Fe concentrations in the overlapping sections of the 2015 and 1992 Gulyia ice cores (i.e., for 1971-1992) are comparable. In the revised version of the paper, the authors indicate (in the Supplement) that "the Al and Fe median concentrations are  $0.3 \mu g/g$  in both records during the 1971–1991 period in which both TE records overlap." However the data are not actually shown. I recommend again that these data should be presented graphically. This would show the reader if the Fe concentrations in the overlapping core sections vary in agreement. The fact that they have the same median concentrations does not necessarily imply that they do. [Response] Done.

The plot with the Fe concentrations of the 2015 and 1992 Guliya ice cores is now shown in Figure S1.

**L187-189: Results of cluster analyses of geochemical data should be treated with a great deal of caution, as they are very sensitive to data pre-treatment (example: transformations) and the choice of the clustering algorithm. See Templ et al. (2008; Applied Geochemistry 23: 2198–2213). The results presented on Fig. S4 would be more convincing if the authors could show that they can be replicated with different clustering criteria or methods. [Response] Done.**

We performed an additional cluster analysis using K-means algorithm (non-hierarchical method recommended in Templ et al. 2008) and the results are shown along with those from the hierarchical cluster analysis in Figure S5. The K-means clustering analysis forms slightly different groups, when compared with the hierarchical analysis but Pb and Zn remain in the same cluster.

We also changed the text (L189-196) to reflect the results from both cluster analyses and to clarify that this is an exploratory data analysis tool as mentioned in Templ et al. 2008. The text now reads: "A hierarchical cluster analysis using the Ward linkage method and the Euclidan distance measure, and a non-hierarchical (K-means) cluster analysis were performed with Factors 1-3 to explore the possible TEs distribution into associated groups. (Fig. S5). Both cluster analyses show that Pb and Zn are strongly associated suggesting these TEs likely have common origins."

On Fig. S1, aluminum (Al) shows consistently negative Excesses (i.e., deficits) in the older Gulyia (1992) ice core, including during the "pre-industrial" interval of reference (1650-1750). This is odd. The concentration of trace metals of crustal origin in environmental matrices (e.g., soils, ice) often have positively skewed probability distributions, but not that of major elements such as Al and Fe. Therefore I would expect the probability distribution of the Al/Fe ratio in the ice to be symmetrical, possibly normal. Hence, if the Ex(Al) shown

on Fig. S1 were calculated as per equation (2) in the paper, I would also expect that in the part of the core that corresponds to the interval of reference (1650-1750) there should be both positive and negative Ex(Al), depending on whether the Al/Fe ratio measured in any part of the core fell above or below the median Al/Fe value for the whole reference interval. In order for the calculated Al Excess to be consistently negative during this interval of reference, the measured Al/Fe ratios must be consistently lower than the reference median Al/Fe value used in the calculation, which necessarily implies that this value can't actually be the median. Something seems to be wrong here, and it may also affect the calculation of Excesses for other trace elements. Maybe there is something missing in the description of the calculation method?

[Response] We reviewed the calculations of the [TE/Fe] median during the pre-industrial period (1650-1750). We had originally obtained the median concentrations of the TEs during the pre-industrial period and then performed the ratio of those values (e.g., [Al]1650-1750 median / [Fe]1650-1750 median). We now did the [TE/Fe] for all individual samples and then obtained the median of the ratios between 1650-1750. The plot has been changed in Fig. S2, and as the reviewer pointed out, the Al excess now show both positive and negative Excess concentration during the 1650-1750 period.

Excess concentrations for all TEs were recalculated for both the 1992 and the 2015 ice cores. Likewise, the statistical analyses were redone with the corrected Excess concentrations. The corrected Excess concentrations are slightly larger than those obtained before, however they do not change either the statistical results or the interpretation.

Fig. 2 has also been updated with the correct Excess concentrations.

Sections 3.1 and Figs. 4-5. The attribution of anthropogenically-derived trace metals deposited in the Gulvia ice core to specific regional sources is largely based on visual "curve matching" of trends between the ice-core composite EF (for Cd, Pb, Zn, and Ni) and in regional emission data. This is fair enough, and I think that there is a good case to be made that the increase in the composite EFs points to dominant sources from East Asia and/or the Indian sub-continent (which is hardly surprising). I am less convinced by the argument offered for the predominance of emissions from specific fossil fuel sources in Pakistan. The observed trend in EFs is probably the result of a mixture of emissions from various regions/sectors. Hence, there could in fact be more than one combination of regional/sector emission sources that could produce the observed trend in EF, but the only such combination that is analyzed in detail is that of emissions from Pakistan (Fig. 5). The argument offered in support of Pakistan would be more convincing if it could be shown that no other combination of regional sources can explain the observed trend in EFs. Ultimately, the "case" for the predominance of Pakistan seems to depend on the apparent "peak" in EFs around 2007, which could match a period of peak emissions in Pakistan at that time (if emission data from this region are to be trusted). Given that the factor analysis attributes only 2 % of the variance in TEs to anthropogenic sources (the rest being associated with crustal sources), the authors should refrain from over-interpreting minor features in the TE record. This is not to say that Pakistan may not be an important source of TEs to the Gulvia ice cap (it would be surprising if it were not), but I think that the relative dominance of emissions from this region is overstated.

[Response] We agree with the reviewer that the EF increasing trend might be a combination of regions and/or sectors, so we have changed the attribution of TEs to all the regions that influence Guliya (South Asia, western China, and Central Asia) throughout the discussion, abstract and conclusions.

Section 3.2.: I find the discussion of the possible influence of the NAO on atmospheric trace element deposition on Gulyia ice cap to be weak. First, the purported correspondence between high/low NAO phases and the composite index of trace metal EF on the Gulyia ice

cap (Fig. 6) is based on a subjective visual comparison, without any supporting qualitative metrics (e.g., correlation coefficients), and it is, to me, unconvincing. Second, this comparison does not offer a definitive way to discriminate or parse, in a quantitative way, the relative influences of the NAO and of anthropogenic source emissions of trace elements, such that one is left to speculate about which factor(s) were dominant at different times. Thirdly, no explicit mechanism is offered in the text to account for this purported relationship. I am assuming the interpretation is the same as that previously suggested in Sierra-Hernandez et al. (2018), i.e. stronger wintertime NAO => enhanced westerlies => more efficient transport of atmospheric trace elements from distant (European) sources in the west to the Gulyia ice cap. As I pointed out in my previous review, this is at odds with the effects of the NAO on atmospheric flow over the Tibetan Plateau in climatology publications (Mao et al., 2011; doi:10.1016/j.atmosenv.2010.10.020; Han et al. 2008; doi:10.1016/j.atmosenv.2007.12.025). Furthermore, there seems to be no clear or consistent association between predominantly low(high) NAO phases and variations in dust deposition on the Gulvia ice cap, as one might expect if the NAO-westerlies linkage was important for atmospheric particulate matter transport (Fig. 6-7 in Thompson et al., 2018). I had previously suggested that one possible way to verify if a stronger winter/spring NAO phase actually enhances east-west atmospheric transport towards the ice cap would be to compare the mean length of air parcel backtrajectories between years of low and high NAO indices. I can only offer the same suggestion again. Unless this or some other supporting evidence can be offered, I recommend that this section be excised altogether from the paper.

[Response] We agree with the reviewer (and the other reviewers) and believe it is better to eliminate Section 3.2 as comparisons between NAO and Guliya EFs are only visual, mechanisms cannot be established due to the complex atmospheric circulation over the Guliya ice cap, the changing emission sources, and the ice core dating uncertainty (1-2 years).

The Tibetan Plateau (TP) atmospheric circulation is influenced by the continental westerlies and the East Asian and South Asian summer monsoons (Schiemann et al., 2009; Yao et al., 2013; Maussion et al., 2014). During winter the westerlies are strong and dominate over the TP. During summer the monsoon alters the atmospheric circulation such that the westerlies weaken and shift northward to ~40-42°N while the northern limit of the monsoon reaches  $34-35^{\circ}N$  (Tian et al., 2007; Schiemann et al., 2009; Maussion et al., 2014). Thus, the Guliya ice cap ( $35^{\circ}17^{\circ}N$ ) is dominated by westerlies during winter but during summer it can experience a combination of monsoonal and westerly flows due to the shift of the westerly jet to the north and the monsoon onset. Due to the location of the Guliya ice cap at the northern limit of the monsoon transition, it is difficult to establish the relationship between the Guliya EF enrichments and NAO.

We performed a running correlation between the winter NAO index and the EF composites of the 1992 and 2015 Guliya core to understand the correlation over time (see Figure 1 below). Unfortunately, with the number of data points we can only do a 5-year running mean and only coefficients > 0.8 are significant (p = 0.05). The NAO-EF relationship is positive before the 1970s and after 2000. However, as the reviewer says these positive visual correlations are not definitive to discriminate or parse, in a quantitative way, the influences of the NAO and those of the anthropogenic sources of TEs especially post-1970 due to the number of emission sources. At least one emission source must emit TEs to detect in the ice core, but given that there is a large number of post-1970 emission sources and they all have different trends, we cannot determine the role that NAO plays in the transport of atmospheric pollutants to the Guliya ice cap.

Figure 1. Five-year running correlation coefficient between the winter NAO index and the 1992 and the 2015 Guliya EF composites. The two horizontal lines show the significance level (p = 0.05).

As suggested by the reviewer, to verify if a stronger winter/spring NAO phase actually enhances east-west atmospheric transport towards the ice cap we looked at air parcel back-trajectories for two years, 1964 (negative NAO) and 1993 (positive NAO) (see Figure 2 below). During 1994 (positive NAO phase), the trajectories seem to go further back to the northwest compared to the 1964 winter when NAO was in a negative phase. This would suggest that the westerly jet position might be influenced by the NAO phase.

---

## Author Response (AR2)

**Co-Editor's comments**

Line 18: "TE" is not defined
[Response] It is defined now "trace element (TE)…".

Line 22: delete "that"
[Response] Deleted.

Line 74: "Inaccurate statistical information"?
[Response] This has been changed to "…inaccurate inventories and emission factors…"

**Reviewer #2 comments**

L20: "likely" is redundant.
[Response] "likely" was remove.

L22-24: The last sentence of the abstract mixes a conclusion ("this new record demonstrates that...") with a policy recommendation ("mitigation policies should be enforced..."). These are separate things. The conclusion belongs in the abstract, the policy recommendation does not.
[Response] Done. The policy recommendation "mitigation policies should be enforced..." was removed.

Introduction: Throughout the text, the words "significant" or "significantly" are used where "large", "strong" or such words would do just as well. It is best to reserve the use of the words "significant" or "significantly" to specific statements about statistical strength.
[Response] This has been changed throughout the text.

L26: "Atmospheric EMISSIONS [or LEVELS] of trace elements (...) have dramatically increased..."
[Response] Changed to "Atmospheric levels of trace elements…"

L59: Maybe "slower" instead of "lower"?
[Response] It has been changed to slower.

L99: I suggest: "Then timescale was constructed by annual layer counting, also constrained by three reference horizons (...)"
[Response] The sentence was changed as suggested.

L149: "...using the upper continental CRUST background..."
[Response] "Crust" was added.

Results and discussion:

Why is the symbol rho used for pairwise correlations in some parts of the text, and the symbol r used elsewhere? It seems inconsistent.
[Response] The Spearman correlation "r" is denoted with rho as opposed to the Pearson's correlation coefficient which is denoted with "r". We have added "Spearman correlations (correlation coefficient $\rho$)" in Sect. 2.4 Statistical Analysis to clarify.

L180: "A factor analysis method was performed ..."
[Response] We are not sure what the Reviewer wanted to suggest here. We changed that part of the sentence to "Factor analysis was used to assess….".

L224: "...and that overprinting DISTURBANCE of the TE records..."
[Response] Disturbance was added as suggested.

L248: This sentence should not begin by "While", but maybe by "Meanwhile, biomass ...".
[Response] "While" was replaced by "Meanwhile,…" as suggested.